



**EXPERIMENTAL AND NUMERICAL INVESTIGATION OF FLOW HYDRAULICS AND**
**PIPE GEOMETRY ON LEAKAGE BEHAVIOR OF LABORATORY WATER NETWORK**
**DISTRIBUTION SYSTEMS**
Tamer Nabil[1*], Fahad Alhaddad[1], Mohamed Dawood[1], Osama Sharaf [1]
*[1] Mechanical Engineering Dept., Faculty of Engineering, Suez Canal University, Ismailia, Egypt*
*[*]Corresponding author: Tamer Nabil*
*Emails: tamer_mtc@yahoo.com, tamir.nabil@eng.suez.edu.eg*
**Abstract**
As the leakage behavior of water distribution network is considered life-threatening and
critical issue, so the behavior of water distribution network system is investigated
experimentally and numerically under the effect of different positions and flow rates of
leakage outlets taking into consideretion the flow hydraulics and pipe geametry. A
laboratory model of the real studied water distribution network is constructed. The
laboratory water distribution network is horizontal and has 16 loops with total length 30 m
and different diameters.The leakage position in the laboratory water distribution network
is altered between main, sub-main and branch pipelines with different flow rates. The
characteristics of the ideal laboratory water distribution network with no-leakage are
studied first. The studied laboratory water distribution network system parameters are
solved theoretically using Hardy-Cross method with seven iterations. The studied water
distribution network system was simulated using computational fluid dynamics technique
Ansys Fluent 18.2. The aim is to modify the ancient water distribution network by sensing
the pressure values using dispersed pressure sensors. Also, from the pressure map of the
laboratory water distribution network, the leakage position if exist can be localized.
Depending on the sensed pressure, the control circuit programmed to close the
corresponding solenoid valves. The leakage flow rates are 0.1, 0.25 and 0.4 L/s and
changed between the main and sub-main pipes. The maximum pressure drop around 500pa
at the node directly preceding the leakage point at leakage flow rate 0.4 L/s. The
performance of the used solenoid valve is simulated using Matlab-Simulink technique. The
simulation results show the response to step down control signal is over damped with
steady state error 2% and settling time 0.6 s.
*Keywords:* Water distribution network, Water leakage, Solenoid valve, Computational
Fluid Dynamics (CFD), Matlab.
**1. INTRODUCTION**
A consistent supply of clean water is the first and most critical community service that
people need. A safe supply of potable water is the basic necessity of mankind; therefore,
water supply systems are the most important public utility (Creaco and Pezzinga, 2018).
The network distribution system is used to supply water from its source to the point of
usage (Ahmad Fuad et al., 2019). The leakage is defined as (amount of) water which
escapes from the pipe network by means other than through a controlled action.



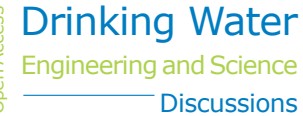

Meniconi et al., 2013 explained the analysis concerning the importance in numerical
models of unsteady friction and viscoelasticity to transients in plastic pipes with an external
flow due to a leak. Tests are based on laboratory experiments, and the use of different
numerical models. Daniel Paluszczyszyna, et al., 2015 Modeled and simulated the water
distribution systems with quantised state system methods (QSS). Daniel Paluszczyszyn et
al., 2015 developed the water network model reduction software. The application can be
integrated with other concepts applied to water distribution system or it can be used as a
standalone tool for the purpose of the model simplification only.
Creaco and Pezzinga, 2018 explored the simulation and optimization modelling, topology
and partitioning, water quality, and service effectiveness. Creaco and Pezzinga, 2014
showed how pipe replacements and control valve installations can be optimized in water
distribution networks to reduce leakage, under minimum nodal pressure constraints.
Santonastaso et al., 2019 proposed a general framework to adjust water distribution
network (WDN) partitioning algorithms to account for the real positions of isolation valves.
Avi Ostfeld, 2015 verified that the water distribution system is a complex assembly of
hydraulic control elements connected together to convey quantities of water from sources
to consumers. Alsharqawi et al., 2020 studied the aging water distribution networks in the
US to verify that they are approaching the end of their useful life, and more than 240,000
pipeline breaks are estimated to occur every year. Starczewska et al., 2015 showed the
common existence of pressure transients in operational water distribution systems requires
their characterization and assessment of their impact by evidencing the occurrence and the
differences in pressure transient behavior in complex WDS.
Del Giudice and Di Cristo, 2003 showed three different sensitivity-based methods for
selecting the worthwhile sensor location in water distribution network. The results show
that there are no marked differences between the three methods. Kumar et al., 2010
presented that the estimation of pipe roughness coefficients is an important task to be
carried out before any water distribution network model is used for online applications.
Galuppini et al., 2019 displayed that real time pressure control is commonly adopted in
water distribution network management to reduce leakage. A numerical description of the
dynamic behavior of the water distribution network (WDN) is introduced. Misiunas et al.,
2006 presented an algorithm for the burst detection and location in water distribution
networks based on the continuous monitoring of the flow rate at the entry point of the
network and the pressure at a number of points within the network.
Fontana et al., 2017 indicated that a common strategy for leakage reduction in water
distribution networks (WDNs) is the use of pressure reducing valves (PRVs). As well
known, a relationship between pressure and water losses can be established, according to
which reducing pressure results in reduced losses. Quraishi and Al-Dhowalia 1994
demonstrated a practical and more reliable approach for assessment of leakage from
Riyadh water distribution network. It presents the methodology and discusses the result of
the field study of ten selected areas of the city. Chiplunkar et al. 1990 Analyzed the looped
distribution networks which is a prerequisite in design or reorganization of water supply
systems. Constantin et al., 2011 reasoned the transient movement results as a hydraulic
system response to sudden valve maneuvers in a water supply network. Numerical and
experimental investigation on pressure variation was carried out.
Choi et al., 2020 claimed the water distribution systems in Korea are responsible for
maintaining a stable supply of tap water and ensuring water quality are experiencing many
problems, such as pipe leakage, corrosion, and aging of pipes. Paez et al., 2018 proposed a
non-iterative method to perform the simulation of water distribution systems with pressure
driven demands using EPANET2 without the need to use its programmer's toolkit. Straka
et al., 2010 studied the distribution networks and their classification and showed that there
is a possible connection between the producers and consumers in two categories. The first
is the economy side, the other side is the production distribution. Latchoomun et al., 2015
proposed a novel model development of old water distribution networks based on
estimation of the leakage from MNF and the burst frequency of AZPs.
Mair et al., 2014 analyzed the impact and effect of improving the data from other sources
for creating water distribution system models. These Investigations showed that hydraulic
WDS models with a mean pressure error of 3m can be created by knowing a percent of
30% of pipes with a diameter ≥250 mm. Athanasios et al., 2009 presented the description
of the technical and physics of the AE leak detection methodology going to its pros and
cons and all the requirements of this technique. Mircea Dobriceanu 2008 performed a
SCADA system for the water distribution stations to monitor and control of the
technological parameters. Konnur et al., 2016 made quick review for the methods of
analyzing and design of multi reservoir multi junction water transmission networks that is
considered to be one of the vital elements for every water supply system.    D'Ambrosio
2015 studied mathematical programming methods in water networks optimization.
Between the major topics they focused on two different and related problems. One
described by the notion of network design, while the other one is more applied in terms of
network operation. Marko Blažević et al., 2005 investigated the various methods of leak
detection in underground network of municipal water distribution system.From the
previous survey and discussed papers, the authors claimed that there is a gap in the leakage
behavior investigation of small water distribution network with the effects of flow
hydraulics and pipe geometry. Also, the gap of the possible treatment ways with leakages
especially in aged water distribution network. The used pressure sensors permit the authors
to figure out the pressure distribution in conventional (ideal) network and network with
leakages. Moreover, the water distribution network pressure map can be drawn in two
cases, ideal with no-leakage and with leakage cases, to standardize the leakage effect. The
network performance is investigated in two cases; at peak hours and off peak hours. The
water distribution network is modified with control circuit to sense the pressure values and
close all pipelines directed to the leakage outlets by solenoid valves if required. Finally,
the study of the water distribution network used theoretical, experimental and numerical
techniques to obtain the behavior of small laboratory water distribution network with and
without leakage effects. Also, the simulation of the used solenoid valve enables the
estimation of the secondary leakage through these valves.

## 2. THEORETICAL BACKGROUND

With respect to distribution network analysis, the conventional theoretical solution is
known as the Hardy-Cross method (Hardy Cross, 1936). The Hardy Cross method is an
iterative technique for equations of flow; continuity of flow (the flow in is equal to the flow
out at each junction) and continuity of potential (the total directional head loss along any
loop in the system is zero) (El-Zahab and Zayed, 2019). The Hardy Cross method depends
on simple mathematics and it iteratively corrects the mistakes in the initial guess used to
solve the problem (self-correcting) (Volokh, 2002). The theoretical results of Hardy Cross
method are obtained after seven iterations and listed in Table 2.

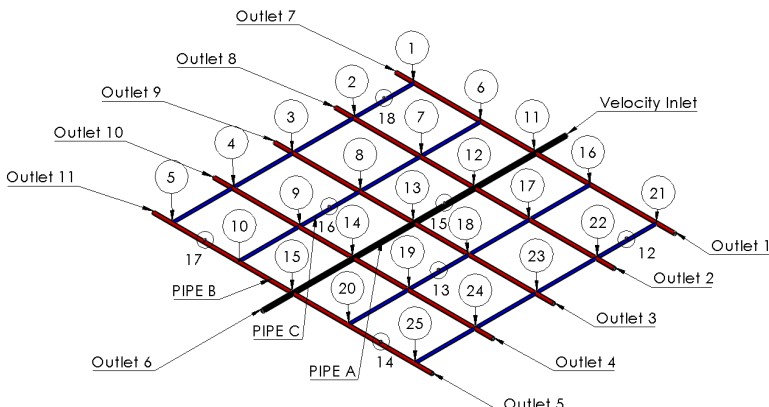

Fig.1 CAD drawing of the laboratory distribution network


**3. EXPERIMENTAL WORK**
Fig.1 shows the CAD drawing and Fig.2 shows the photo of the laboratory water
distribution network. The studied water distribution network consists of main, sub-main,
branch pipelines and branching nodes that creating 16 loops. The main pipeline has 1inch
diameter and 3m length, the sub-mains have 0.75inch diameter and 12m total length and
the branches have 0.5inch diameter and 15m total length. Ball valves are used in the
laboratory water distribution network to control the flow rate and direction.

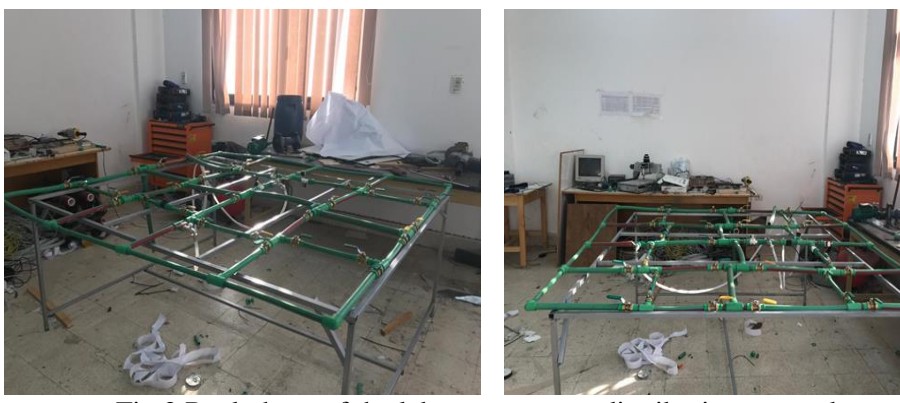

Fig.2 Real photo of the laboratory water distribution network

Fig.3 shows the ball valve which installed to discharge the water with different flow rates
exit from the water distribution network as a leakage. The leakage ball valve is placed in
the main and sub-main pipelines to investigate experimentally the most critical leakage
cases in the water distribution network analysis.

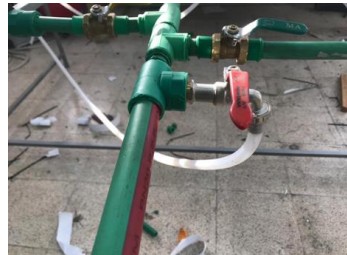

Fig.3 Photo of the ball valve at the main pipeline to simulate the leakage outlet

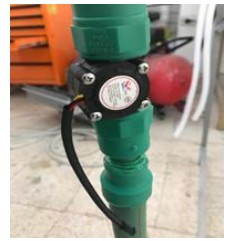
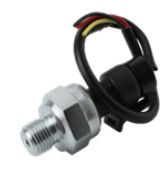
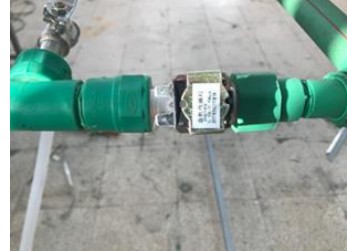

(a) Flow meter sensor      (b) Pressure sensor      (c) Solenoid valve

Fig.4 Photos of sensors and valve of the laboratory water distribution network

Table 1. Specifications of flow meter sensor, pressure sensor and solenoid valve

| Flow meter sensor (Sea YF-S201) *(www.seagroup.com)* | Pressure sensor (Flying Elephant SE0006) *(www.flyingelephant.lk)* | Solenoid valve (Adafruit ADA997) *(www.adafruit.com)* |
|---|---|---|
| Range:1-30 L/min Water pressure: ≤1.75 MPa Working voltage range: DC 5～18 V Load capacity: ≤10 mA(DC 5V)Operating temperature range: ≤80℃ Operating humidity range: 35%～90%RH (no frost) | Supply voltage:5.0VDC Output: 0.5~4.5VDC Working current: ≤10mA Pressure range: 0~1.2MPa Proof Pressure: 2.4MPa Burst Pressure: 3.0MPa Op. Temp.: 0~85℃ Storage Temp.: 0~100℃ Measure error: ±1.5%FSO Full temp. range error: ±3.5%FSO Response: ≤0.9S | 1/2" nominal NPS Working pressure of fluid: 0.02-0.8 Mpa Working temperature of fluid: 1ºC - 100ºC Voltage: 6VDC to 12VDC Current: 500mA Materials: Stainless Steel/Poly-oxy-methylene Operating mode: Normally closed Filter Screen: Stainless Steel Inlet Filter Usage: Water |


The measured pressure values via pressure sensors are transferred instantaneously to be
recorded on the computer to give a real picture of the pressure map through the whole
network. One flow meter is fixed and changed between the main, sub-main and branch
pipelines to measure the flow rates (Fuad et al., 2019). A control unit is designed with code
programmed using the Arduino for a specified function. As the pressure sensors sense the
pressure drop to a certain setting value, the signal from control unit initiate the solenoid



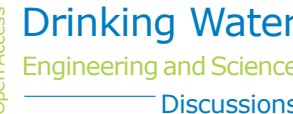

valves to close all pipelines which directed the flow to the leakage outlet. Meanwhile, this
technique is important to save considerable amounts of water losses. The studied laboratory
water distribution network meets all the requirements of the model (geometrically,
kinematically and dynamically similar) except it constructed horizontally in the laboratory
with no potential head and neglected hydraulic gradient energy.
Fig.4 and Table 1 illustrate the components and their specifications of the studied laboratory
water distribution network. The flow is discharged from water tank to the water distribution
network by ½ hp centrifugal pump (H=18m, Q=22L/min and 2850rpm).

### 3.1. control circuit and algorithm

Pressure measurements are recorded by pressure sensors through Arduino Mega. When the
leakage is started, this causes a pressure drop and the controller determines the location of
the leakage and defines the pipe number from the node where the pressure is decreased.
Then controller sends a signal to close all the solenoid valves in pipelines which directs the
water flow to leakage location. Also, an alarm is sent to the related person about the leakage
location. Fig.5 shows the flow chart of the control algorith. Fig.6 shows the schematic of
wiring and connections of the control circuit components and the interaction communicating
signals between these components. These two figures summarize the used methodology of
the studied experimental work.

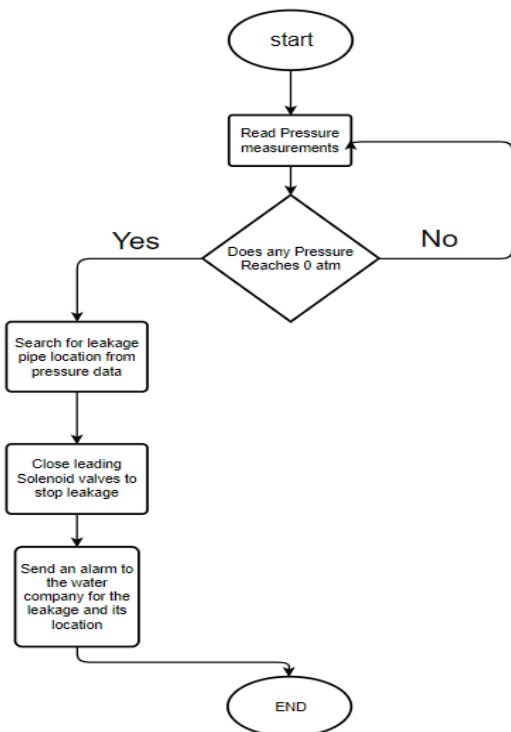

Fig.5 Flow chart of Arduino control algorithm to control the leakage by solenoid
valves using pressure signals


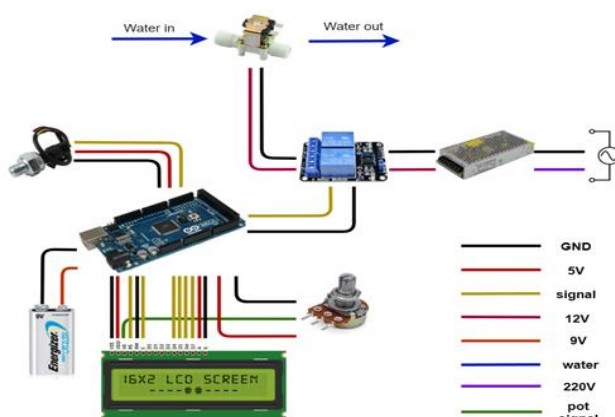

Fig.6 Schematic drawing of the network control circuit


The control circuit consists of; relay module (2 channel), Arduino Mega 2560 R3, relay
driver module RK4, Power distributer, LCD and Volt Current monitor gauge DSN-VC288.
**3.2. Experimental Procedures**
In small scale network like that under consideration, the friction is high and consequently
the losses are also high and this is a general trend in small pipes. Also, all the fittings,
elbows and other minor losses sources have a considerable effect. The most influenced
pipelines of the network due to leakage must be determined, hence the leakage has different
impact on the pipelines of the network. The response of the solenoid valve to the control
signal command must be investigated.
The experimental test is performed for conventional laboratory water distribution network
without leakage. The pressure and flow rate values in different nodes and pipelines are
recorded as reference values. Open the ball valve which discharges the water out of
network as a simulation of leakage. The throttle area of the ball valve is adjusted to certain
leakage flow rate by measuring the water volume in calibrated container and stop watch to
measure the time. After adjusting the leakage flow rate, the solenoid valves at leakage
position are observed to determine its speed of response. The leakage flow rate is changed
and the values of flow rates and pressures are recorded and the response of the solenoid
valves are detected ( Ferrante et al., 2012).
**4. NUMERICAL SIMULATIONS**
The numerical simulations are performed in case of no leakage of the water distribution
network to enable the detection of the ideal behaviour of the network, as a first case (Loan
Sarbu, 2010). After that, the computations are performed with leakages at different positions
in the network, as a second case (Chandapillai et al., 2011).
The numerical model was mapped on a commercial CFD solver Ansys fluent 18.2.
Parametric studies of velocity, pressure and eddy viscosity profiles are investigated. Also,
vector display of water flow along the different locations of the network (pipes and nodes)
gives a detailed picture of the flow in the network. The Phase coupled SIMPLE (PC-



SIMPLE) algorithm was used for the pressure-velocity coupling discretization while the
body force is used for pressure discretization (Afifi et al., 2018).
The initial conditions were; Uniform fully developed velocity profile at pipe inlet. First-
order upwind discretization scheme was used for the momentum equations, turbulence
kinetic energy (k), and turbulence dissipation rate (ε). All the iterative solutions were
performed in double precisions. An inlet flow rate boundary condition was used at the pipe
inlet (Greyvenstein and Van Zyl, 2007). Using flow rate at inlet (0.1, 0.25 and 0.4 L/s) and
pressure at outlet (atmospheric pressure) as boundary conditions are the common way of
formulating pipe network flow problems. The usual no-slip boundary condition was
adopted at the pipe wall. To avoid divergence, under-relaxation technique was applied. The
under-relaxation factor for pressure was 0.3, for momentum was 0.6, and these for
turbulence kinetic energy and its dispassion rate were 0.8. The solution was assumed to
have converged when the continuity and velocity residuals reached nearly $10^{-4}$ which is a
promising value in the solution according to Ansys fluent manual. The numerical solution
typically required 480 iterations (Van Zyl and Clayton, 2007).
The water distribution network consists of 1" pipe with inner diameter of 26.24 mm and
length 2500mm which is the main pipe, 0.75" pipe with inner diameter of 20.57 mm and
length 1000mm which are the sub-main pipes, 0.5" pipe with inner diameter of 15.47 mm
and length 500mm which are the branches pipes and for leakage a 12 mm inner diameter
pipe was assumed as a source of leakage. All these pipes and their fittings are PVC.
Fig.7 shows the locations of the leakage in the water distribution network. The number of
leakage outlets is 7. One leakage position at the main pipeline (number 15), two leakage
positions at the sub-main pipelines (number 14 and 17, symmetric positions), two leakage
positions at external branches pipelines (number 12 and 18, symmetric positions) and two
leakage positions at internal branches pipelines (numbers 13 and 16, symmetric positions).

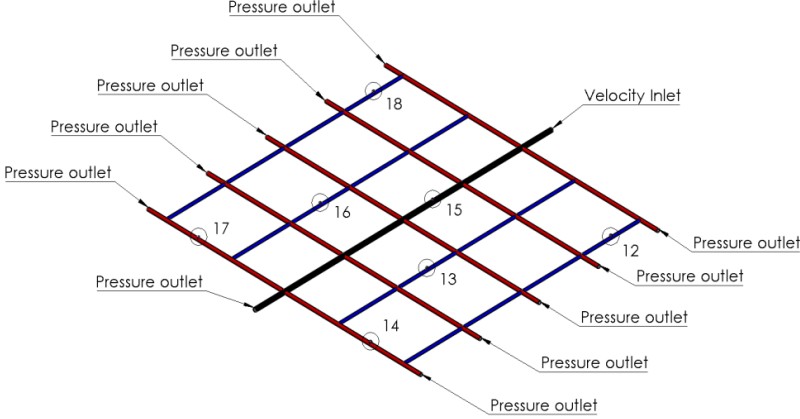

Fig.7 Water distribution network leakage outlet locations

Fig.7 also shows the distribution network boundary conditions. The boundary conditions
are assumed to have an inlet velocity at 2 m/s, and outlets with atmospheric pressure. All
turbulence parameters were set to 1% turbulent intensity with equivalent hydraulic
diameter.



Fig.8 demonstrates that the meshing was done in ICEM CFD using blocking technique
with one million Hexa elements, with angle quality criteria greater than 27 and determinant
2x2x2 quality criteria greater than 0.3 which are acceptable according to ICEM CFD user
manual for Fluent solver (http://www.ansys.com, 2013).

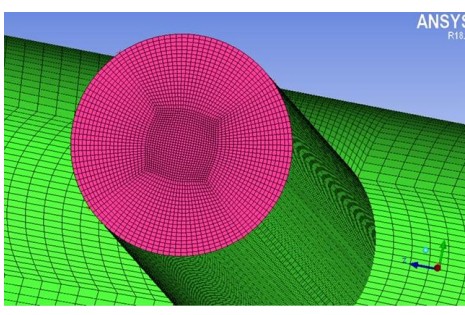
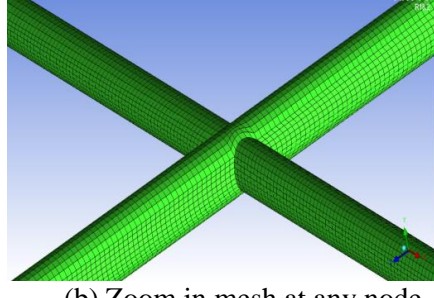

(a) Leakage outlet mesh                    (b) Zoom in mesh at any node
Fig.8 Distribution network mesh configuration

The mesh grid stability test was carried out in order to insure accurate simulation results
with precision and to save the required simulation time. The model was tested using three
different mesh numbers to obtain mesh grid stability and variables steadiness at the
optimum mesh grid number. The pressure value at different nodes is the chosen parameter
to determine the required mesh number. Fig.9 shows that 1 million elements are acceptable
number of meshes due to the changed variables (node pressure) become constant at this
number (Brki and Praks, 2019).

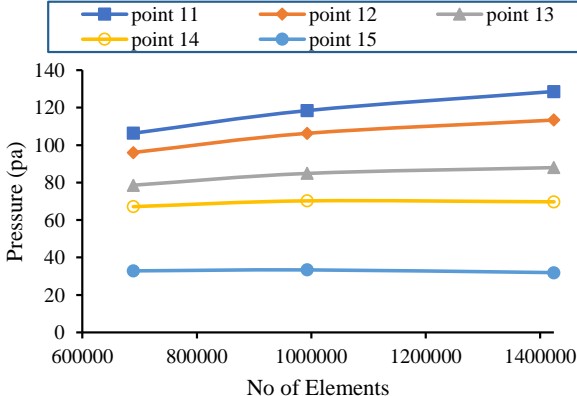

Fig.9 pressure variations with mesh grid numbers at network nodes 11-15

The software Ansys fluent was used for solving the governing equations. This package
utilizes a method of control volume theory to convert the governing equations to algebraic
equations so that they can be illuminated numerically. The governing equations (continuity,
momentum and energy) associated with the standard k-ε model (Turbulent kinetic energy
equation, Dissipation rate equation) using the default values of the empirical constants
(Launder and Spalding, 1974).
Table 2 shows the theoretical (Hardy-Cross using Darcy-Wesibach method) and numerical
(Ansys, Fluent) results in case of ideal (simulated by allowing all outlets from 1 to 11



opened) with no leakage in the water distribution network. The results show an agreement
between the theoretical and numerical results. The theoretical calculation is an iterative
method with 7 steps of iterations until the head losses in the studied loop tends to zero but
the number of iterations in the numerical calculations equal 480 iterations (Brki´c, 2016).

Table 2. Theoretical and numerical flow rates at different loci of network

| Outlet no/Flow rate | Theoretical (Kg/s) | Numerical (Kg/s) |
|---|---|---|
| 1 | 0.1718 | 0.16182 |
| 2 | 0.1365 | 0.12084 |
| 3 | 0.0919 | 0.07143 |
| 4 | 0.0510 | 0.04405 |
| 5 | 0.0398 | 0.03052 |
| 6 | 0.3779 | 0.48447 |
| 7 | 0.1718 | 0.16182 |
| 8 | 0.1365 | 0.12084 |
| 9 | 0.0919 | 0.07143 |
| 10 | 0.0510 | 0.04405 |
| 11 | 0.0398 | 0.03052 |


## 257 5. Experimental error and uncertainty analysis

The precision of the recorded experimental results must be investigated to be confirmed. The
measuring devices must be calibrated primarily then their uncertainty can be estimated depending
on their accuracy and dependent errors.
The uncertainty analysis can be calculated and depends on the error of direct measured data.
$$U_P = \sqrt{\left(\frac{\partial P}{\partial x} \times U_x\right)^2 + \left(\frac{\partial P}{\partial y} \times U_y\right)^2} \tag{1}$$

Where; $U_P$ is the uncertainty of the studied parameter (P) which depends on the variable parameters
(x, y) and the errors of these variables ($U_x$, $U_y$). Kline and McClintock, 1953 showed the
experimental measuring data has certain error range depending on the accuracy of the measuring
device. The measured data are water pressure, water flow rate, response time of the solenoid valve.
The accuracies for used devices are listed in Table 3 according to the manufacturer.
Table 3. Uncertainty for all measuring devices.

| Device | Model | Accuracy | Range | Error |
|---|---|---|---|---|
| Flowmeter Sensor | Sea: YF-S201 | ± 0.1 L/min | 1 - 30 L/min | 1.8 % |
| Pressure Sensor | Flying Elephant :SE0006 | ± 0.01 MPa | 0 - 1.2 MPa | 1.5% |
| Solenoid Valve | Adafruit :ADA997 | Response time (open): ≤ 0.15 sec Response time (close): ≤ 0.3 sec | Pressure: 0.02-0.8 Mpa Temperature: 1-100 ºC Voltage: 6-12VDC | 2% |
| Calibrated flask | - | ± 20 ml | 0 – 5000 ml | 4 % |


## 270 6. RESULTS AND DISCUSSIONS



Firstly, the numerical results should be validated by experimental results at different leakage flow rates of different nodes in case of leakage in main pipeline of the network. Secondly, the numerical results of the network performance and flow velocity vectors at all nodes under different leakage positions (sub-main and main) and flow rates are investigated. Finally, the performance of the solenoid valve and its time step response are investigated.

Fig.10 shows the experimental and numerical pressure results at nodes from 16 to 20 of the distribution network in case of leakage at outlet 15 located in the main pipeline. The pressures at nodes from 16 to 20 are recorded numerically and measured experimentally at different leakage flow rates at outlet 15. Experimental and numerical results show the pressure values at nodes from 16 to 20 decrease due to the leakage flow rate increase. The difference in pressure values between the experimental and numerical have different ranges according to the nodes position. Node 18 is the highest affected pressures node, also at this node the difference between the numerical and experimental results ranges from 13% to 21%. Node 18 location is in the nearest pipeline parallel to the main pipeline which contain the leakage outlet. Nodes 16, 17, 19 and 20 nearly have tiny differences between the experimental pressure results and numerical pressure results.

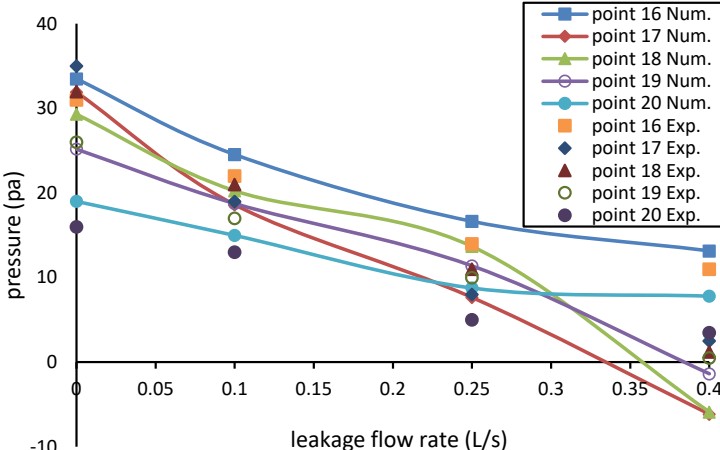

Fig.10 Validation for the numerical code with experimental work for the pressure at nodes (16-20) versus outlet 15 leakage flow rate variations




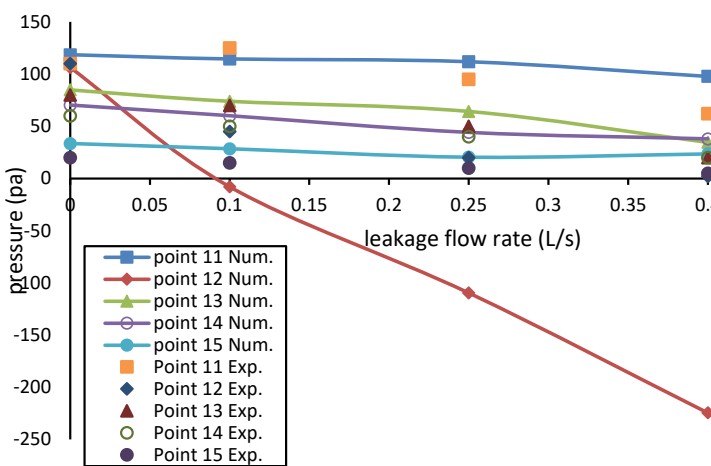

Fig.11 Validation for the numerical code with experimental work for the pressure at
nodes (11-15) versus outlet 15 leakage flow rate variations

Fig.11 shows the experimental and numerical pressure results at nodes from 11 to 15 of the
distribution network in case of leakage at outlet 15 in the main pipeline. The pressures at
nodes from 11 to 15 which are located along the main pipeline of the distribution network
are recorded numerically and measured experimentally at different leakage flow rates at
outlet 15. Node 13 is the highest pressure drop influenced node due to leakage flow rate
variations at the main pipeline of the distribution network, also at this node the difference
between the numerical and experimental results ranges from 9% to 25%. Node 13 location
is aligned with outlet leakage 15 and is the nearest node to this outlet.

## 6.1. CFD Simulation due to Leakage at Sub-Main

Outlet 14 of leakage outlets is located at the sub-main pipeline of the water distribution
network which is perpendicular to the main pipeline and between nodes 20 and 25. The
leakage flow rate at outlet 14 can be varied to study its response on the network behavior
by measuring the pressure at different network nodes. These leakage flow rate variations
enable us to draw a map of pressure variations in the network to locate the most affected
regions and record the different network effects with this leakage position.

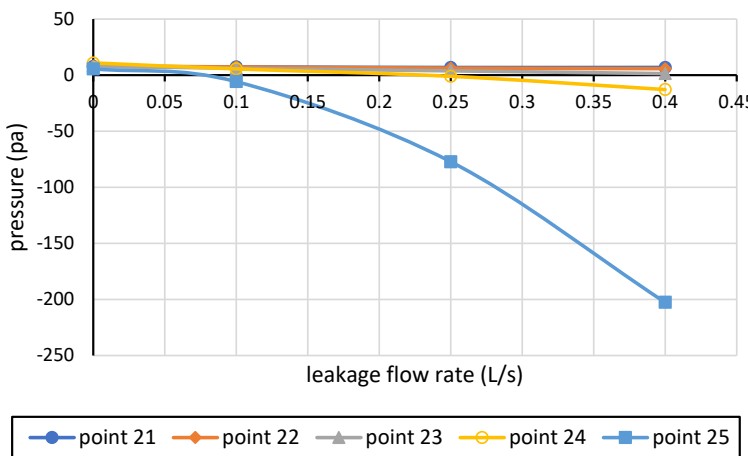

Fig.12 Pressure at nodes (21-25) versus outlet 14 leakage flow rate variations

Fig.12 shows the pressure variation at nodes from 21 to 25 with leakage flow rate variation
at leakage outlet 14. As the leakage flow rate at outlet 14 increases the pressure at node 25
extremely influenced and decreases sharply especially at high leakage flow rate because
this node located at the start or the end (according to the flow directions) of the
corresponding sub-main containing the leakage outlet. Nodes 21, 22, 23 and 24 have small
pressure variation effect with leakage flow rate variation at leakage outlet 14. Pressure at
node 25 is highly affected due to this node is a corner node of the closed loop that contain
the leakage location.

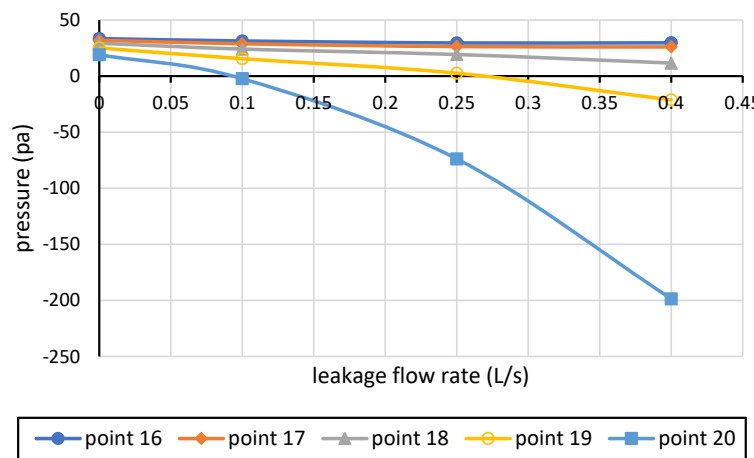

Fig.13 Pressure at nodes (16-20) versus outlet 14 leakage flow rate variations

Fig.13 shows the pressure variation at nodes from 16 to 20 with leakage flow rate variation
at leakage outlet 14. As the leakage flow rate at outlet 14 increases the pressure at node 20
decreases sharply especially at high leakage flow rate because this node is the other end
(with node 25) of the sub-main pipeline containing the leakage outlet. Nodes 16, 17, 18



**Drinking Water**
Engineering and Science
——————— Discussions

and 19 have small pressure variation effect with leakage flow rate variation at leakage
outlet 14.

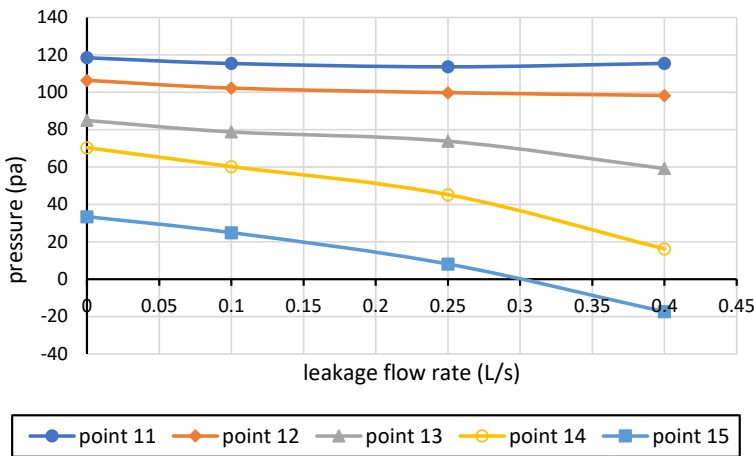

Fig.14 Pressure at nodes (11-15) versus outlet 14 leakage flow rate variations

Fig.14 shows the pressure variation at nodes from 11 to 15 with leakage flow rate variation
at outlet 14. The nodes from 11 to 15 are located at the main pipeline of the distribution
network. As the leakage flow rate at leakage outlet 14 increases the pressure at nodes 11
and 12 nearly constant. The pressure at nodes 13, 14 and 15 decrease with same trend as
the leakage flow rate increase. So, the leakage flow rate variation at outlet leak point in the
sub-main pipeline has apparent effect on the pressure in the main pipeline especially at
nodes located at the end of the main pipeline.

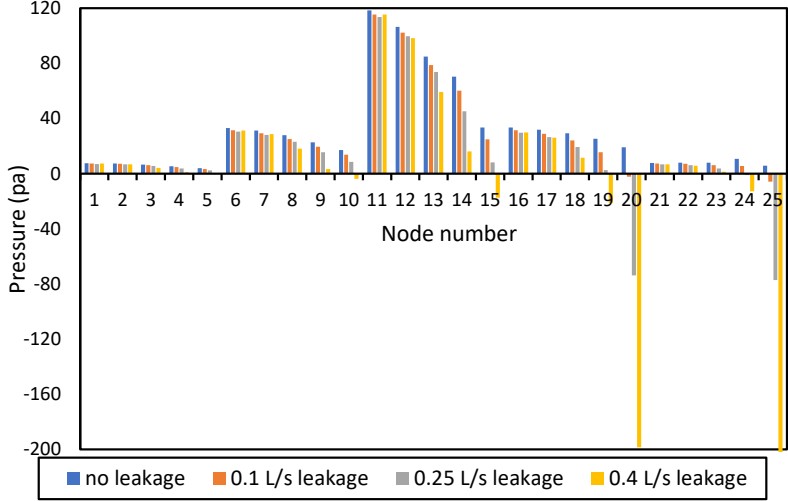

Fig.15 Histogram of pressure variations at all network nodes due to leakage flow
rate variations at outlet 14

Fig.15 is the histogram of the pressure variations at all water distribution network nodes as
a result of leakage flow rate variations at designed leakage outlet 14 at the external sub-



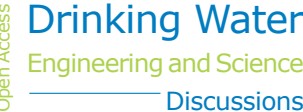

main of the distribution network. From the analysis of the pressure variations, the effect is
observable in all nodes at alignment with the corresponding sub-main pipeline containing
the leakage outlet 14. The effect is decreased in nodes located in line parallel to the
corresponding sub-main pipeline, as the distance from sub-main pipeline increases the
leakage effect decreases. Also, the effect is noticeable at nodes located as apex in the closed
loops that contain the designed leakage outlet 14.
These figures indicate that each node has been affected with different performance
according to the leakage position and the leakage flow rate. The negative pressure at nodes
20 and 25 indicates the pressure values at these nodes are vacuum pressures and the air
bubbles are forms at these nodes which influence the streaming of the flow results in the
decreasing of the flow area. Also, the negative pressure causes back flow and flow
separation at these nodes as a leakage which apparent in velocity vector contours.
Fig.16 shows the velocity vectors at different nodes of the distribution network in case of
leakage at outlet 14 in the external sub-main pipeline of the network. The studied pipeline
alignment is in x-axis direction which is perpendicular to the main pipeline. The maximum
flow rate in laboratory distribution network is 0.4 L/s. Figs.16 (a) and (b) illustrate the
velocity vectors of the fluid flow at nodes 20 and 25. Nodes 20 and 25 are the extreme
nodes of the chosen sub-main pipeline of the network containing the leakage outlet 14.
Figs.16 (a) and (b) show that the flow direction in this pipeline is directed from nodes 20
and 25 to the leakage outlet 14 due to the sudden drop in pressure value that results at
leakage outlet. Also, this pipeline feeds another loop by water in negative x-axis direction
through node 20 but due to leakage the flow is reversed in opposite direction (positive x-
axis) which causes trouble-shooting at this loop. So, in this case, two solenoid valves are
initiated to close the pipeline discharges the flow to the leakage location. Fig.16 (d) show
the velocity vector at node 19. At this node 19 the flow separation and vorticity is appeared
specially at sharp edges which considered as a considerable leakage value. Fig.16 (c) show
the velocity vector at node 14 in the main pipeline. Node 14 velocity vector demonstrates
that the maximum flow at this pipeline and the flow direction in negative Z-axis direction,
also the vorticity, circulation and separation occurs at nodes aligned with the main pipeline.

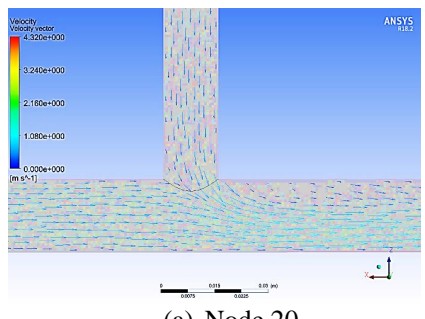

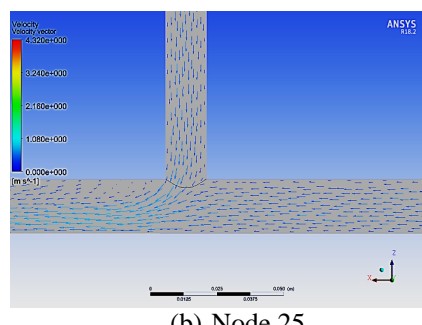

|  (a)  Node 20  |  (b)  Node 25  |

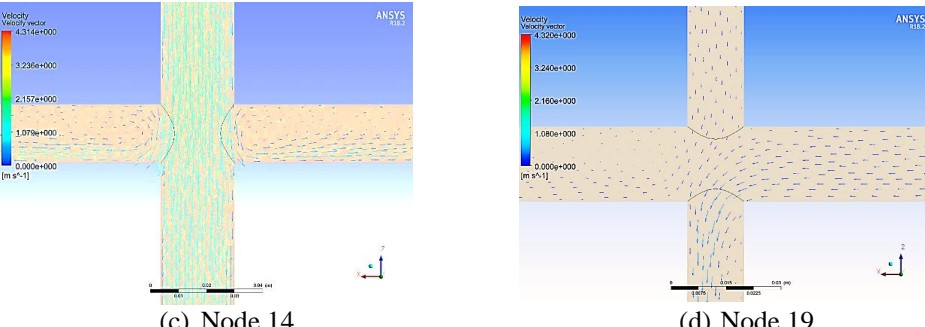

(c) Node 14      (d) Node 19

Fig.16 Velocity vectors on different nodes with leakage flow rate 0.4L/s at outlet 14

## 6.2. CFD Simulation of Leakage at Main Pipeline

Leakage outlet 15 is located in the leakage design framework at the main pipeline of the
water distribution network between nodes 12 and 13. This case is critical because the
leakage in the main pipeline causes noticeable change in the network behavior and
influences the consumptions everywhere in the network. These leakage flow rate variations
enable us to draw a map of pressure variations in the network at different conditions to
locate the most affected regions.

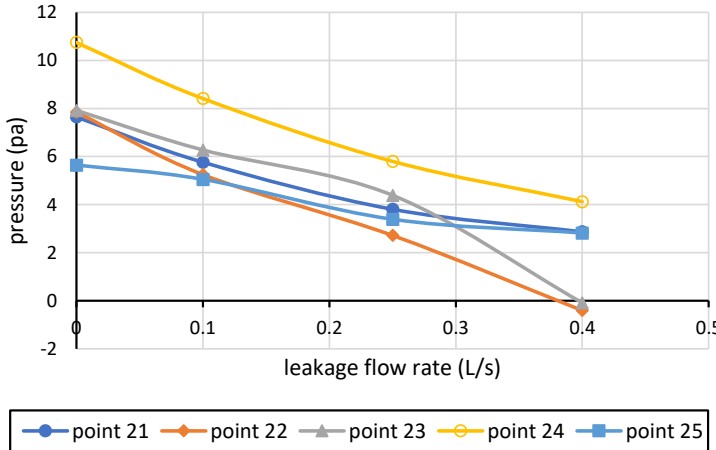

Fig.17 Pressure at nodes (21-25) versus outlet 15 leakage flow rate variations

Fig.17 shows the pressure variation at nodes from 21 to 25 with leakage flow rate variation
at leakage outlet 15. Outlet 15 is located at the main pipeline between nodes 12 and 13. As
the leakage flow rate at leakage outlet 15 increases the pressure at nodes 22, 23 and 24
decrease to low values but at high leakage flow rate the slope of pressure variation of node
23 higher than other two nodes 22 and 24. The pressure variations curves at nodes 21 and
25 with outlet 15 leakage flow rate variations have small variations.
Fig.18 shows the pressure variation at nodes from 16 to 20 with leakage flow rate variation
at leakage outlet 15. As the leakage flow rate at outlet 15 increases the pressure at nodes
17, 18 and 19 decrease to low values with different trends. At low leakage flow rate, point
17 has greater slope and its pressure decreases sharply with leakage flow rate variation. At
high leakage flow rate, node 18 has greater slope and its pressure highly decreases as



leakage flow rate increases. The pressure values at nodes 16 and 20 due to leakage flow
rate variations at outlet 15 have small variations.

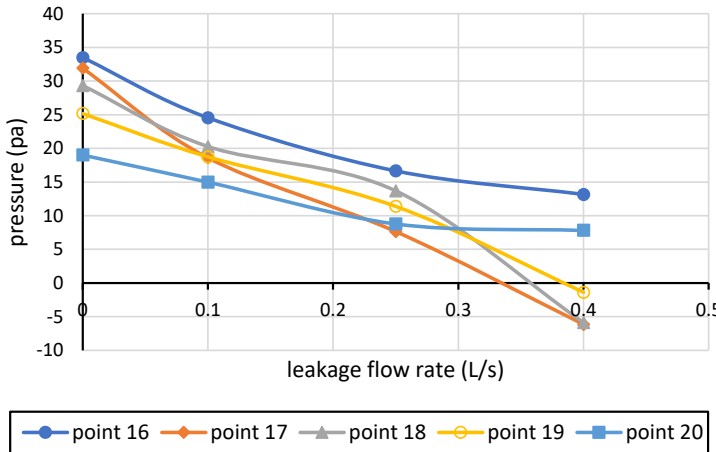

Fig.18 Pressure at nodes (16-20) versus outlet 15 leakage flow rate variations

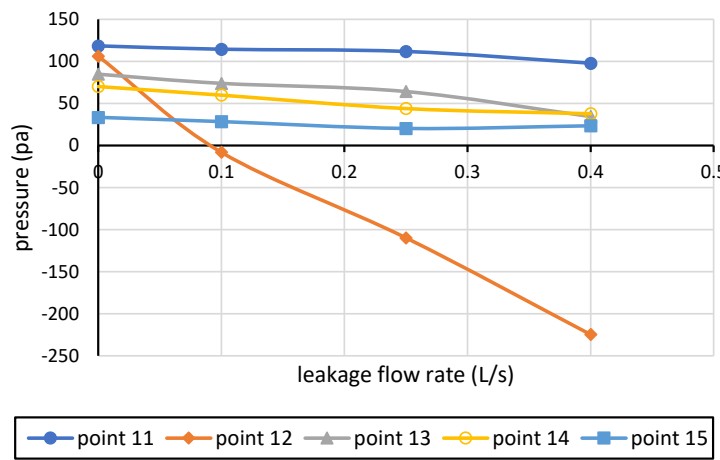

Fig.19 Pressure at nodes (11-15) versus outlet 15 leakage flow rate variations

Fig.19 shows the pressure variation at nodes from 11 to 15 with leakage flow rate variation
at leakage outlet 15. The nodes from 11 to 15 are located at the main pipeline. As the
leakage flow rate at outlet 15 increases the pressure at node 12 decrease sharply to low
values. The pressure variations at nodes 11, 13, 14 and 15 with leakage outlet 15 flow rate
variations have small variations.



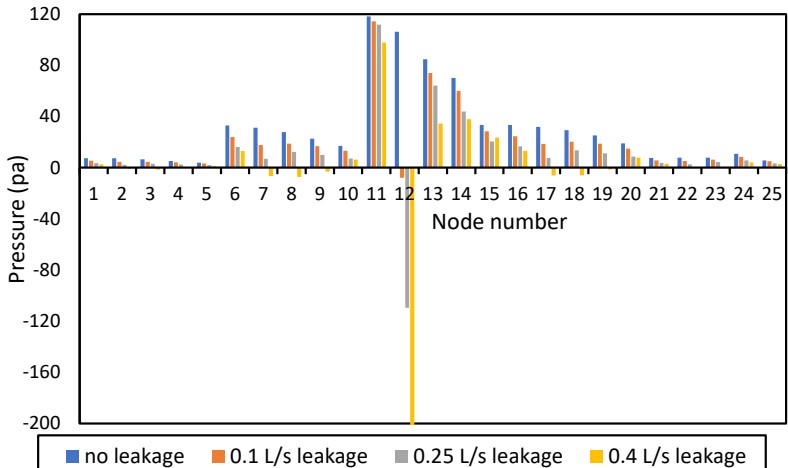

Fig.20 Histogram of pressure variations at all network nodes due to leakage flow
rate variations at outlet 15

Fig.20 the histogram shows the pressure variations at all nodes as a result of leakage flow
rate variations at designed leakage outlet 15. The leakage effect is obvious in all extreme
nodes of the branches parallel to the part of main pipeline containing the leakage outlet 15.
The effect is noticeable at corner nodes of the two loops that contain the designed leakage
outlet 15 in a pipeline considered as a common pipe between these loops due to leakage.
Fig.21 shows the velocity vectors at different nodes of the distribution network in case of
leakage at outlet 15 in the main pipeline of the network. Figs.21 (c) and (d) illustrate the
velocity vectors of the fluid flow at nodes 12 and 13. Nodes 12 and 13 are the extreme
nodes of the main pipeline of the network containing the leakage outlet 15. Figs.21 (c) and
(d) show that the flow direction in this main pipeline is unidirectional from node 12 to node
13 so that the pressure drop due to leakage in this main pipeline has no effect on the flow
direction. So, in modification the distribution network to save the water due to leakage one
solenoid valves preceding the leakage position on the main pipeline must be used to close
the direction to the leakage location. Figs.21 (a) and (b) show the velocity vector at node
17 and 7 in the closest pipeline parallel to the main pipeline containing the leakage. Also,
the flow is directed to these nodes to circulate in the loops and finally directed to the
leakage outlet due to this is considered the lowest pressure in the network. At these nodes
7 and 17 the flow separation and vorticity are appeared specially at sharp edges as a
distribution node which are significant in total leakage calculations and have symmetric
flow configurations.

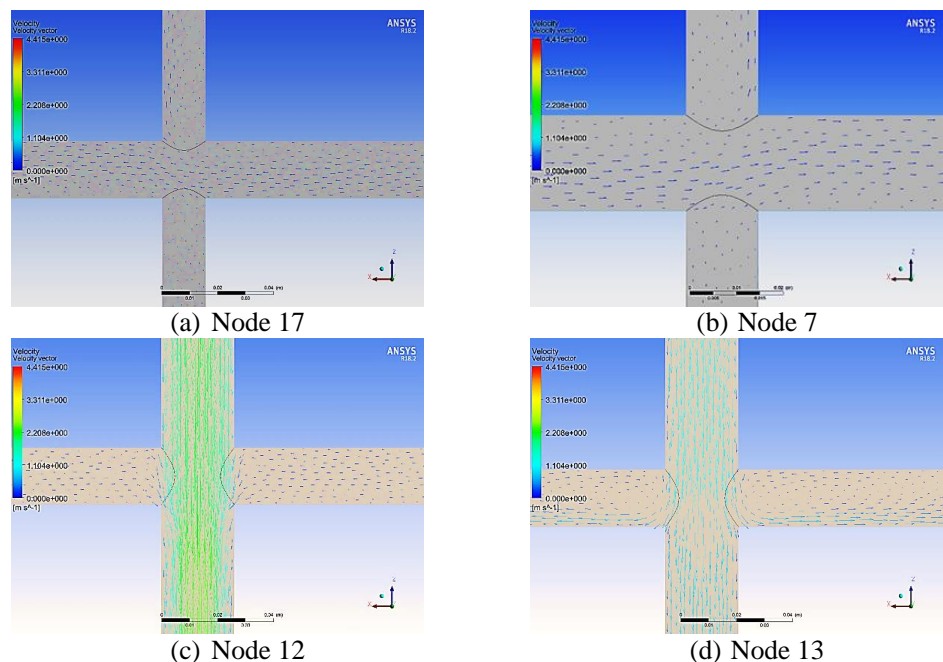

| (a) Node 17 | (b) Node 7 |
| (c) Node 12 | (d) Node 13 |

Fig.21 Velocity vectors at different network nodes at leakage flow rate 0.4L/s at outlet 15

## 7. Matlab Simulation of Solenoid Valve

Simulink is a program for simulating any system depending on the derived mathematical
model of that system like the integrated distribution network system or any component of
the network as the solenoid valve. The solenoid valve actuated by the control circuit due
to leakage to close the pipeline directed the flow to the leakage position. The valve
actuation response time is investigated due to this response time is a main factor in total
leakage calculation. The leakage flow will directly affect the pressure response
characteristics of the solenoid valve (Southern, 2016).
Leakage during and after the solenoid valve closing reduces the efficiency and changes the
performance of the water distribution network control. The leakage in the solenoid valve
is mainly caused by the clearance between the plunger and the casing of the valve at the
end of the valve operation, due to the delay of valve time response and the elongation of
closing time.
The solenoid valve performance is investigated by deriving the mathematical model of the
valve (non-linear differential equations). The deduced mathematical model is used to
develop computer simulation program for the studied valve by Matlab-Simulink. The
mathematical model consists of continuity equation, flow rate equations and equation of
motion. The valve time response is studied under the effect of step pressure signal (Kabib
et al., 2016).
The continuity equation in the valve chamber can be represented as:

$$Q_i - Q_o - A_f \frac{dx}{dt} - \frac{V+Ax}{B} \frac{dP}{dt} = 0 \qquad (2)$$

The valve water inlet flow rate can be calculated as follow:



$$Q_i = C_d A_i \sqrt{\frac{2}{\rho}(P_p - P_i)} \qquad (3)$$

The valve water exit flow rate can be calculated as follow:

$$Q_e = C_d A_e \sqrt{\frac{2}{\rho}(P_e - P_a)} \qquad (4)$$

The moving parts can be moved under the action of pressure forces, magnetic force, spring
force, inertia force, viscous force and limiting force.

$$P_i A_f - P_e A_b + F_m - F_L = m\frac{d^2x}{dt^2} + f\frac{dx}{dt} + k(x + x_o) \qquad (5)$$

The moving part is limited mechanically by the valve body material and a counter reaction
force is developed as:

$$F_L = \begin{cases} |x|K_L + f_L \dfrac{dx}{dt} & x \le 0 \\ 0 & x > 0 \end{cases} \qquad (6)$$

The magnetic force depends on magnetic field intensity and the magnetic resistance as:

$$F_m = \frac{1}{2}\phi_{air}^2 \frac{1}{\mu_o A} \qquad (7)$$


Fig.22 displays the pressure response of the water flow through the solenoid valve due to
the step drop of pressure in the main pipeline due to leakage in this line. The simulation
shows the step response of the solenoid valve integrated in the distribution network along
the main pipeline as the flow pressure in the main pipeline step decreased from 1.8 bar to
zero bar (gauge pressure). The solenoid valve in the main pipeline closes immediately the
way of flow to the leakage outlet as the drop of pressure is recorded.
Fig.22 shows that the exit water pressure from the solenoid valve reaches a steady state
value equal to the leakage pressure. The response showed over damped high oscillation
with a small steady state error about 2%. The steady state error indicates that the valve did
not close completely and the leakage still remained but with small amounts. The results
show also the settling time of the valve is nearly 0.6 sec. This time is taken into
consideration when calculating the total leakage amount of water.

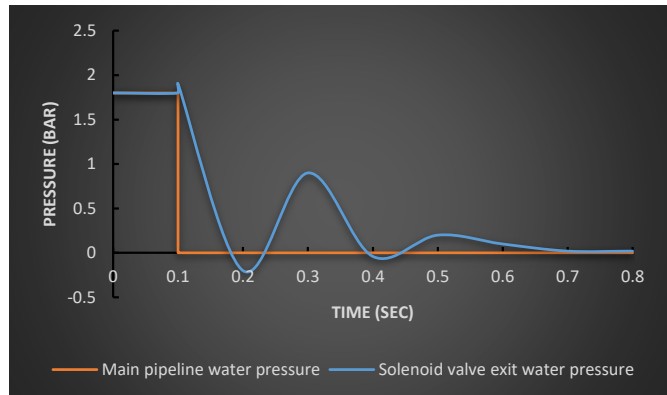

Fig.22 Solenoid valve response to main pipeline water pressure step down variation

**8. CONCLUSION**



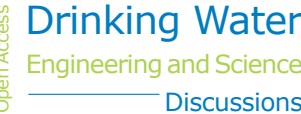

This paper has proposed a new technique of investigation the water leakage effects on the
water distribution network performance. This investigation was performed experimentally,
theoretically and numerically. There is an agreement between the experimental and
numerical results with error range from 9% to 25% according to the distance and the path
from the leakage outlet to the effected nodes. Theoretical calculations used Hardy-Cross
method with seven iterations. Numerical simulation uses Ansys Fluent 18.2 has a benefit
as good approach of this study with 480 iterations. The leakage flow rates have values 0.1,
0.25 and 0.4 L/s and changed between the main and sub-main pipes. The research work is
limited to small values of leakage flow rate due to the small dimensions of the laboratory
network. The maximum pressure drop was 500pa at the node directly preceding the leakage
outlet at the leakage flow rate 0.4 L/s of the main pipeline. The most influenced nodes are
that near to the leakage outlet. The leakage at the main pipeline in the water distribution
network is considered the most critical leakage case. The flow direction was reversed and
the flow separated in certain leakage cases. The pressure sensors sensed the pressure values
and sent signals continually to the control circuit. The control circuit according to the
programmed control algorithm gives orders to close the pipelines directed the flow to the
leakage outlet by solenoid valves. The cost, the solenoid valves and pressure sensors
fixation with linkage to the control circuit are considered the difficulties that can be
encountered when implementing this method in a real world situation. Aaccording to the
nodes pressure drop and their positions the accurate loci of leakage can be determined. The
performance of the used solenoid valve is simulated using Matlab-Simulink technique. The
simulation results show the valve response to step down pressure control signal is over
damped high oscillatory with a small steady state error 2% and settling time 0.6 sec.
*Nomenclature*

| | |
|---|---|
| $A_f$ | Valve plunger face area (m$^2$) |
| $A_b$ | Valve plunger back area (m$^2$) |
| $A_i$ | Valve inlet orifice area (m$^2$) |
| $A_e$ | Valve exit orifice area (m$^2$) |
| B | Bulk modulus of water (pa) |
| $C_d$ | Discharge coefficient |
| $F_L$ | Valve moving part limiting force (N) |
| $F_m$ | Magnetic force (N) |
| f | Friction coefficient (N.s/m) |
| $f_L$ | Limiter damping coefficient (N.s/m) |
| k | Spring stiffness (N/m) |
| $k_L$ | Limiter material stiffness (N/m) |
| m | Valve moving part mass (kg) |
| P | Valve chamber pressure (pa) |
| $P_p$ | Pump pressure (pa) |
| $P_i$ | Valve inlet pressure (pa) |
| $P_e$ | Valve exit pressure (pa) |
| $P_a$ | Atmospheric pressure (pa) |
| $Q_i$ | Valve inlet flow rate (L/s) |
| $Q_o$ | Valve outlet flow rate (L/s) |



x      Valve plunger displacement (m)
$x_o$      Spring pre-compression length (m)
V      Valve Chamber initial volume ($m^3$)
$\phi_{air}$      Magnetic flux of air gab (V.s)
$\mu_o$      Permeability of vacuum ($N/A^2$)

*Acronyms*
(WDN) Water Distribution Network
(CFD) Computational Fluid Dynamics
(PVC) Polyvinyl Chloride
(CAD) Computer Aided Design
(PRV) Pressure Reducing Valve

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

Methods Appl. Mech. Eng. 3, 1974, 269−289.
D. Brki´c, Spreadsheet-based pipe networks analysis for teaching and learning purpose.
Spreadsheets    Educ.    (EJSIE),    2016,    9,    4646.    Available    online:
https://sie.scholasticahq.com/article/4646.pdf.
S.J. Kline, F.A. McClintock, Describing uncertainties in single sample experiments. Mech. Eng.
75, 1953, 3–8.
Jacob D. Southern, Nonlinear Time-Frequency Control of Electromagnetic Solenoid Valve,
ASME International Mechanical Engineering Congress and Exposition, 2016.
M. Kabib,  I. Made, L. Batan,  A. Sigit, Modelling and simulation analysis of solenoid valve for
spring constant influence to dynamic response, Journal of Engineering and Applied Sciences
561      11(4):2790, 2016, 2790-2793.