# Peer review of "EXPERIMENTAL AND NUMERICAL INVESTIGATION OF FLOW HYDRAULICS AND PIPE GEOMETRY ON LEAKAGE BEHAVIOR OF LABORATORY WATER NETWORK DISTRIBUTION SYSTEMS"

_Drinking Water Engineering and Science, 2020_

## Short Comment (SC1) · 8 Jun 2020

This manuscript is important to figure out and investigate the water distribution network prototype performance under studying the effect of varying the leakage position and flow rate on the performance of small similitude laboratory water distribution network.

---

## Referee Comment (RC1) · Anonymous Referee #1 · 20 Jul 2020

Abstract: it is really not clear what the purpose of the work is, what the scientific contribution is and how the proposed methods are chosen.

- The English needs to improve significantly, as there are too many to list structural, grammatical and typological errors. The language used in the paper is a bit 'flowery', starting from the first sentence of the abstract. The abstract and so the paper should be restructured to clearly motivate what the engineering/societal objective is; there is a large set of literature on leakage detection using various methods, which can be references to clearly set the context. Then the choices of experiments and method

should be justified.

Brevity and unsuggestive language is more appropriate when discussing health risks associated with microbial contamination. I suggest a review of language used, typos, and grammar.

- it is really not clear what such a scaled experiment adds, the claim that WDN real leakage tests are difficult and so building a lab model brings more value is not valid. In fact, leak detection and localisation methods with distributed sensors in the real network and with simulation of the real system are successfully applied in practice (as demonstrated by numerous manuscripts in literature. - why is a hardy cross method used when there are free, modern alternatives that even allow pressure driven leakage simulations in milliseconds (eg. EPANET, WNTR, etc) - it is not clear what these are: ancient WDN, dispersed sensors, the pressure numbers, etc? - the paper seems to mix up leak detection using transient methods with study state approaches using pressure sensors, and control of solenoid valves. - what is really the proposed 'new technique' of investigation the water leakage effects on the water distribution network performance; what is the performance index? What is the leakage localisation method in Figure 5.

Although the paper may potentially have some useful contributions, the structure and quality of the writing makes it difficult to grasp what the contributions are, what has been done and why, and what the relationship is with the rich recent literature on leakage diagnosis and localisation.

---

## Referee Comment (RC2) · Anonymous Referee #2 · 21 Jul 2020

The authors performed an experimental and numerical investigation on leakage behaviour, influenced by pipe geometry and hydraulics. Rewording would be to improve the quality of the paper and make it easier to read. For instance, the Introduction section is difficult to follow/read. It should be re-written to be less general and to provide more specific information directly linked to the objectives of the paper. Sections 2 (theoretical background), 3 (experimental work), 4 (numerical simulations), 5 (experimental error and uncertainty analysis) and one part of the section 7 (Matlab simulation of solenoid valve) should be combined into one section (Materials and methods). This

section should be written so that the reader has a better understanding of how the research was performed. The number of Figures and Tables should be reduced both in Materials and methods and Results sections. When the results are presented, less general comments should be made and specific comparisons with the literature should be made. Overall: The idea of the paper is interesting, but the authors should consider re-writing it to meet the standards of DWES Journal.

Specific comments are given below:

Line 12 How is leakage behaviour considered life-threatening?

Line 15: Spelling errors: consideretion and geametry. There are other typos further in the text. Therefore, general spelling and grammar check is also essential.

Line 24 - 25 The sentence is not clear, please rephrase. What do you mean by ancient?

Line 31- 33 The sentence is not clear, please rephrase.

Line 38 – 118 – As given in general comments, the Introduction section should be re-written in order to be less general and to provide more specific information directly linked to the objectives of the paper.

Line 136: The Figure belongs to Experimental work.

Section 6, first paragraph (there are no text lines that I could refer to): This is summary of your methodology, not results.

Line 145: Combine Fig 2, 3, 4 into one Figure.

Line 150: Table 1 is better suited for Supplementary information.

Line 165: Use a capital letter for the sub-section title.

Line 228: Combine Figure 1 and Figure 7 into one.

Line 236 – 256: This is too detailed, please put it into Supplementary information.

[Figure]

Line 257: You have introduced Experimental error and uncertainty analysis. However, the uncertainty was not discussed in the paper; and you only briefly discussed the experimental error. Please provide more info regarding this.

Line 268: Table 3 is better suited for Supplementary information, but the error should be discussed in the paper.

Line 270 – 384: As given in general comments, the number of Figures and Tables should be reduced. When the results are presented, less general comments should be made and specific comparisons with the literature should be made.

Line 271: What do you mean by validation in Figure caption? It looks more like a comparison. Also, combine Figure 10 and Figure 11 into one.

Line 272: It is evident from the Figure 11, that experimental and numerical results for the point 12 are considerably different. However, this was not explained in the text. Please include an explanation here.

Line 287: combine Figure 12 and Figure 13 into one (also further in the paper do the same).

Line 385 – 410 – These lines belong to the Material and method section.

Line 421: What do you mean by small?

Line 422 - 423: Have you calculated "leakage amount of water"? If yes, please specify.

Line 426 – 448 – This is more summary and observations, rather than conclusions. Please specify what you can conclude from your observations (results).

Line 442 – 444 –First you include/discuss this in the discussion part, and only then you can put it into conclusions.

---

## Author Comment (AC1) · 5 Aug 2020

Response to Interactive Comments of Anonymous Referee #1 I'd like to thank you so much for your guidance, you gave me motivation to work and you supported my ideas and results. Also your questions and notices offer me a brainstorming way to solve them. Thank you for your encouragement advices and suggestions, I really appreciate your precious time you took to help me with this and in the following my response to your comments.

[Figure]

Abstract: it is really not clear what the purpose of the work is, what the scientific con-
tribution is and how the proposed methods are chosen. The abstract is adapted to
clear: (1) The purposes: 1- Investigate the effect of leakage on the different nodal
pressure in the network 2- Determination the most affected nodes due to the leakage
3- Studying the effect of varying the location of leakage and its flow rate 4- Adapt the
system to close all possible pipelines connect the floe to the leakage position 5- In-
vestigate the response of the solenoid valve (2) The scientific contribution: figure out
the leakage effect on the performance of the water distribution network and scientific
estimation of the leakage effect according its position and rate. (3) The method: due to
the difficult application on the real network so a similitude model is designed and fab-
ricated to facilitate the experimental investigation of the leakage effect, also the com-
putational investigation enables further understanding. The English needs to improve
significantly, as there are too many to list structural, grammatical and typological errors.
The language used in the paper is a bit 'flowery', starting from the first sentence of the
abstract. The abstract and so the paper should be restructured to clearly motivate
what the engineering/societal objective is; there is a large set of literature on leakage
detection using various methods, which can be references to clearly set the context.
Then the choices of experiments and method should be justified. The manuscript is
edited for proper English, grammar, punctuation, spelling by one or two of the highly
qualified native English speakers at the Cambridge Proofreading Group (the certifica-
tion of proofreading is attached). Also, the introduction become more specific to the
paper objective. The paper is restructured to be clear by the sections from 2 to 7 are
combined into one section under title materials and methods. Brevity and unsuggested
language is more appropriate when discussing health risks associated with microbial
contamination. Done I suggest a review of language used, typos,and grammar. The
manuscript is edited for proper English, grammar, punctuation, spelling by one or two
of the highly qualified native English speakers at the Cambridge Proofreading Group
(the certification of proofreading is attached). it is really not clear what such a scaled
experiment adds, the claim that WDN real leakage tests are difficult and so building a

lab model brings more value is not valid. due to the difficult and impossible application of the leakage study on the real water distribution network so a particular similitude model with accurate geometric and dynamic with kinematic similarities is designed and fabricated to facilitate the experimental investigation of the leakage effect, even it gives approximate results it considered good conclusion.

In fact, leak detection and localisation methods with distributed sensors in the real network and with simulation of the real system are successfully applied in practice (as demonstrated by numerous manuscripts in literature. - why is a hardy cross method used when there are free, modern alternatives that even allow pressure driven leakage simulations in milliseconds (eg. EPANET, WNTR, etc). Hardy-Cross method is just a theoretical method to calculate the flow rates in the pipelines which is used to estimate the flow rates in the pipelines before the study. After the theoretical calculations, the experimental work emphasizes the theoretical results. Finally the CFD technique is used instead of EPANET which give accurate simulations in seconds.

it is not clear what these are: ancient WDN, dispersed sensors, the pressure numbers, etc? These words and sentences are edited to be clear Ancient mean old, dispersed mean distributed. Numbers mean values. the paper seems to mix up leak detection using transient methods with study state approaches using pressure sensors, and control of solenoid valves. - what is really the proposed 'new technique' of investigation the water leakage effects on the water distribution network performance. The study in steady state only New is deleted What is the leakage localisation method in Figure 5. The leakage localization was determined in all possible and different configuration, geometry, position, alignment and dimensions such that the pipeline parallel or perpendicular, exterior or interior, main or sub-main. Although the paper may potentially have some useful contributions, the structure and quality of the writing makes it difficult to grasp what the contributions are, what has been done and why, and what the relationship is with the rich recent literature on leakage diagnosis and localisation. The manuscript is edited for proper English, grammar, punctuation, spelling by one or two

of the highly qualified native English speakers at the Cambridge Proofreading Group (the certification of proofreading is attached). Also, the introduction become more specific to the paper objective and the structure is adapted.

Please also note the supplement to this comment:
https://dwes.copernicus.org/preprints/dwes-2020-17/dwes-2020-17-AC1-supplement.pdf

---

## Author Comment (AC2) · 5 Aug 2020

Response to Interactive Comments of Anonymous Referee #2

I'd like to thank you so much for your guidance, you gave me motivation to work and you supported my ideas and results. Also your questions and notices offer me a brainstorming way to solve them. Thank you for your encouragement advices and suggestions, I really appreciate your precious time you took to help me with this and in the following my response to your comments.

[Figure]

The authors performed an experimental and numerical investigation on leakage behavior, influenced by pipe geometry and hydraulics.

Q1. Rewording would be to improve the quality of the paper and make it easier to read. For instance, the Introduction section is difficult to follow/read. It should be re-written to be less general and to provide more specific information directly linked to the objectives of the paper.

The manuscript is edited for proper English, grammar, punctuation, spelling by one or two of the highly qualified native English speakers at the Cambridge Proofreading Group (the certification of proofreading is attached). Also, the introduction become more specific to the paper objective.

Q2. Sections 2 (theoretical background), 3 (experimental work), 4 (numerical simulations), 5 (experimental error and uncertainty analysis) and one part of the section 7 (Matlab simulation of solenoid valve) should be combined into one section (Materials and methods). This section should be written so that the reader has a better understanding of how the research was performed.

Sections from 2 to 7 are combined into one section under title materials and methods.

Q3. The number of Figures and Tables should be reduced both in Materials and methods and Results sections.

The number of figures are reduced

Q4. When the results are presented, less general comments should be made and specific comparisons with the literature should be made. Less general comments are made.

The obtained results of this research are matched with good agreement with literature results in phenomenon and observation and conclusions but the mathematical values are difficult to be compared due to the differences of application.

Q5. Overall: The idea of the paper is interesting, but the authors should consider re-writing it to meet the standards of DWES Journal. Specific comments are given below:

The manuscript has proof reading taking into consideration the standard of DWES.

Q6. Line 12 How is leakage behaviour considered life-threatening?

In this sentence the authors want to show the disadvantage of the leakage in the water distribution network such that it causes a great loss of the limited amount of the available drinking water in the world and in case of great leakage it possible to cause life threatening of peoples.

Q7. Line 15: Spelling errors: consideretion and geametry. There are other typos further in the text. Therefore, general spelling and grammar check is also essential.

The manuscript is edited for proper English, grammar, punctuation, spelling by one or two of the highly qualified native English speakers at the Cambridge Proofreading Group (the certification of proofreading is attached).

Q8. Line 24 - 25 The sentence is not clear, please rephrase. What do you mean by ancient?

Ancient is replaced by old to explain that one of the manuscript applications on the pipeline network constructed from several years with old technology and in the present can't withstand the loads. The sentence is rephrased.

Q9. Line 31- 33 The sentence is not clear, please rephrase. The sentence is rephrased.

Q10. Line 38 – 118 – As given in general comments, the Introduction section should be re-written in order to be less general and to provide more specific information directly linked to the objectives of the paper. The introduction become more specific to the paper objective

Q11. Line 136: The Figure belongs to Experimental work. The figure is added to the

experimental work section

Q12. Section 6, first paragraph (there are no text lines that I could refer to): This is summary of your methodology, not results. This paragraph is re-written in methodology section and deleted from results.

Q13. Line 145: Combine Fig 2, 3, 4 into one Figure. Figures 2, 3 and 4 are combined in figure 2

Q14. Line 150: Table 1 is better suited for Supplementary information. Table 1 is very important to the readers of the manuscript and journal because it is the specifications of the used flow meter sensor, pressure sensor, and solenoid valve in the experimental work

Q15. Line 165: Use a capital letter for the sub-section title. Done

Q16. Line 228: Combine Figure 1 and Figure 7 into one. Done

Q17. Line 236 – 256: This is too detailed, please put it into Supplementary information. This section explains the mesh grid stability of the CFD and it reduced and shortened in two lines.

Q18. Line 257: You have introduced Experimental error and uncertainty analysis. However, the uncertainty was not discussed in the paper; and you only briefly discussed the experimental error. Please provide more info regarding this. The uncertainty words are eliminated due to it didn't studied

Q19. Line 268: Table 3 is better suited for Supplementary information, but the error should be discussed in the paper. Table 3 only shows the accuracy, range and error of the used devices

Q20. Line 270 – 384: As given in general comments, the number of Figures and Tables should be reduced. When the results are presented, less general comments should be made and specific comparisons with the literature should be made. Done

Q21. Line 271: What do you mean by validation in Figure caption? It looks more like a comparison. Also, combine Figure 10 and Figure 11 into one. Validation is replaced by comparison Figs 10 and 11 are the responses of nodes in different pipelines with different response so they are separated.

Q22. Line 272: It is evident from the Figure 11, that experimental and numerical results for the point 12 are considerably different. However, this was not explained in the text. Please include an explanation here. The explanation is included in the text as (Node 12 was the highest pressure-drop affected node due to the leakage flow rate variations at the main pipeline of the distribution network. Also at this node the differences between the numerical and experimental results are great due to, in experimental test the flow pressure is supported from flows of different loops to substitute the pressure drop but numerically this phenomenon can't be simulated and the mathematical equations calculate and iterate the negative pressure drop. Node 12 location was aligned with outlet leakage 15 and was the nearest node to this outlet).

Q23. Line 287: combine Figure 12 and Figure 13 into one (also further in the paper do the same). It is preferred to separate the response of nodes at different pipelines because the difference is obvious if the pipeline parallel or perpendicular, exterior or interior, main or submain so it should be separated.

Q24. Line 385 – 410 – These lines belong to the Material and method section. Done

Q25. Line 421: What do you mean by small? Small means very very tiny leakage amounts but with long time it will be effective

Q26. Line 422 - 423: Have you calculated "leakage amount of water"? If yes, please specify. No, the leakage didn't calculate.

Q27. Line 426 – 448 – This is more summary and observations, rather than conclusions. Please specify what you can conclude from your observations (results). The conclusion is modified to be specific.

Q28. Line 442 – 444 –First you include/discuss this in the discussion part, and only then you can put it into conclusions. This part is removed from conclusion.

Please also note the supplement to this comment:
https://dwes.copernicus.org/preprints/dwes-2020-17/dwes-2020-17-AC2-supplement.pdf